# Flour on Gluten-Free Muffins from Different Edible Cassava Varieties in Thailand

**DOI:** 10.3390/foods11244053

**Published:** 2022-12-15

**Authors:** Rungthip Sangpueak, Chanon Saengchan, Kansinee Laemchiab, Dusadee Kiddeejing, Supatcharee Siriwong, Kanjana Thumanu, Nguyen Huy Hoang, Piyaporn Phansak, Kumrai Buensanteai

**Affiliations:** 1School of Crop Production Technology, Institute of Agricultural Technology, Suranaree University of Technology, Nakhon Ratchasima 30000, Thailand; 2Synchrotron Light Research Institute (Public Organization), Nakhon Ratchasima 30000, Thailand; 3Division of Biology, Faculty of Science, Nakhon Phanom University, Nakhon Phanom 48000, Thailand

**Keywords:** physico-chemical, cassava flour, starch, functional properties, cassava

## Abstract

In Thailand new edible cassava varieties have been developed to be used in the food industry. The aim of this research was to analyze the difference between flour from three cassava varieties and to evaluate the suitability and quality of flour for gluten-free muffins. The physico-chemical properties of flour from three varieties were studied. The results showed the moisture content of flour was between 10.65 ± 0.01 and 10.85 ± 0.45%. Total protein content was highly significant with a difference of 1.97 ± 0.00%, 2.15 ± 0.01%, and 2.18 ± 0.01%, respectively. Moreover, ash and fat in each flour were highly significant. Amylose content was 19.93 ± 0.47%, and the viscosity was 6286.00 ± 1.52 mPa.s. The color of flour values of *L* a* b** value was not statistically different in each variety of flour. Fourier transform infrared spectroscopy (FTIR) analysis was used for the biochemical change in flour. The PCA and cluster analysis results revealed that cassava flour from Pirun 6 was different from Pirun 2 and Pirun 4. After that, the test using selected cassava flour from Pirun 6 to test the physical properties and sensory attributes of gluten-free muffins compared with wheat flour found that gluten-free muffins were overall better than basic muffins.

## 1. Introduction

Cassava (*Manihot esculenta*) is one of the most important economic crops in Thailand with an annual production of around 25 million tons. It is the fifth edible plant, following rice, potato, maize, and wheat [1,2] Cassava is grown throughout the country and can raise income for producers, processors, and traders. Moreover, cassava is used as animal feed and in the manufacture of different industrial products such as starch, modified starch, sweeteners, and derivatives for food and non-food applications [3,4]. Cassava flour is dried cassava in powder and is a product derived from cassava roots. It is used as a food, a stabilizer in many products, and as a thickening agent. It has the potential to replace wheat, which is in high demand. However, there are still few varieties suitable for cassava flour production and their plantation areas in Thailand are limited. The differences in cassava varieties have been reported to play essential roles in flour production and have significantly affected cassava flour’s physico-chemical, functional, and other quality characteristics [5,6,7]. Therefore, the effect of the composition on the quality of cassava flour includes cultivar, the geographical location, the maturity stage of the plant, environmental factors (light, temperature, humidity, water, and nutrition) and environmental conditions pre/post-harvest [6,8].

The main advantage of cassava flour is it is naturally gluten-free and a great flour to use in baking and cooking. So, people who avoid gluten can use it as a replacement for wheat flour in terms of taste and texture. Cassava flour is recommended in the diet of celiac patients who strictly adhere to gluten-free food [9,10]. Cassava flour is used as a raw material to produce gluten-free food products because of the low amount of protein, fiber, fat contents, and hypoallergenic properties [9,11]. One of the most popular uses of cassava flour is as a replacement for wheat flour for bakery applications because of its special quality attributes and substitute for wheat flour in a variety of recipes [12]. However, bakery products from cassava flour still have an unfavorable appearance and crumb texture in terms of softness and springiness of baked products and taste have not been accepted by customers. Therefore, this study aimed to evaluate the suitability and quality of flour for gluten-free muffins from three edible cassava varieties in Thailand.

## 2. Materials and Methods

### 2.1. Raw Materials

Three varieties of edible local cassava (Pirun 2, Pirun 4, and Pirun 6) from field trials grown in organic systems were obtained from Pak Chong district, Nakhon Ratchasima province, Thailand. The cassava was planted in April 2021 (summer season, average temperature 29 °C, average relative humidity 82%) and harvested in December 2021 (winter season, average temperature 24 °C, average relative humidity 66%). The experimental areas were thoroughly plowed two times and high beds of approximately 45 cm were made. Then, planted was completed vertically with 1 × 1 m spacing with a drip irrigation system. The weeds were managed at 1, 2, and 3 months after planting. Harvest was undertaken at 8 months old to produce flour in the next step.

### 2.2. Cassava Flour Preparation

The samples were produced using 50 kg of freshly harvested cassava tubers from each of the varieties that were peeled to remove the outer skin. Then they were soaked in brine and washed to remove stains and dirt. After that, they were cut into 1 mm pieces, then baked (Binder^®^ FD115, Neckarsulm, Baden-Württemberg, Germany) at 60 °C for 48 h, and then ground until fine to obtain powder granules 9–20 microns using a beater (FRITSCH^®^ Cross Beater Mill PULVERISETTE 16, Idar-Oberstein, Rheinland-Pfalz, Germany) for analysis.

### 2.3. The Chemical Properties of Flours from Cassava Varieties

#### 2.3.1. Proximate Analysis

Total moisture content, total ash content, total carbohydrate, fat, and total protein were analyzed by Central Laboratory (Thailand) Co., Ltd. Bangkok, Thailand using an in-house method based on AOAC Official Method 994.12 (2000). The nitrogen content of the samples was determined by the Micro-Kjeldahl method (conversion factor of 5.95) to convert to crude protein. The weight difference methods were used to determine moisture and ash content, while crude fat was determined using the AOAC procedure with petroleum ether as the solvent. The carbohydrate content was calculated by the percentage of difference as Equation (1) [13]:%CHO = 100 − (% moisture + % ash + % lipid + % protein)(1)

#### 2.3.2. Total Cyanide Content, Acrylamide, Amylose, Gluten Content, and Viscosity

The cassava flour samples (100 g) for total cyanide content were determined using the pyridine/pyrazolone method [14]. Acrylamide was determined using the in-house validation of an LC-MS/MS method [15]. Amylose content was determined using the Starch-iodine blue method [16]. Gluten content was performed using a Gliadin/Gluten Biotech commercial ELISA kit [17] and viscosity was determined using RVA Viscosity Parameters (RVA-4, Newport Scientific, Warriewood NSW, Sydney, Australia) [18]. All of the parameters were analyzed by Central Laboratory (Thailand) Co., Ltd. Bangkok, Thailand. The methods were determined using AOAC Official Method 994.12.

#### 2.3.3. Color of Cassava Flour

The color of cassava flour was determined using a Colorimeter (CR-400, Konica Minolta Sensing Inc., Osaka, Japan). Results were expressed in the CIE *L***a** *b** color space.

#### 2.3.4. FTIR Spectrometric Analysis

Cassava flour samples were analyzed to compare the differences of cassava flour from each cassava variety by FTIR. The resolution of flour samples was 4 cm^−1^ and scanned 64 times in the area of 4000–400 cm^−1^. The spectra were corrected with FTIR spectrometer (Tensor 27 from Bruker Optics, Ettlingen, Germany). Spectral equipment control was carried out by OPUS 7.2 software (Bruker Optics Ltd., Ettlingen, Germany) at the Synchrotron Light Research Institute (SLRI), Thailand.

#### 2.3.5. Microstructure of Cassava Flour

Dehydrated cassava flour samples were on a carbon tape attached to a piece of aluminum stubs (Sputtering Device, Balzers Union Limited, Liechtenstein), coated with 35 nm of gold-aluminum, and then observed under a Scanning electron microscope (SEM, FEI Quanta-450; FEI Company, Eindhoven, The Netherlands) at an accelerating voltage of 20 kV and photographed at a magnification of 8000×.

### 2.4. Effects on the Quality and Sensory Characteristics of the Gluten-Free Muffin

#### 2.4.1. Gluten-Free Muffin and All-Purpose Flour Muffin Preparation

All ingredients were weighed according to muffin-making formulations. Using 2 eggs mixed with 100 g of sugar and 1/4 teaspoon (tsp) of salt, the eggs were beaten until they turned pale yellow. Then 125 mg of fresh milk, 60 mg of oil, and 1 tsp of vanilla flavor were added and mixed well. After that, 150 g of cassava flour or all-purpose flour and baking powder 1 ½ tsp were added. The mixture was poured into a muffin cup and baked at 170 °C for 20 min in an electric multideck baking oven (Clarte’ FOHM-20VN, Thailand). A gluten-free muffin from cassava flour was compared with a basic muffin from all-purpose flour in the same proportion.

#### 2.4.2. Bake Loss of the Gluten-Free Muffin and the All-Purpose Flour Muffin

Fifteen weight measurements of gluten-free muffin and all-purpose flour muffin /samples were taken for each formulation before and after baking. Losses were calculated as the following Equation (2):Baking loss % = 100 − (100 × weight after baking/weight before baking)(2)

#### 2.4.3. Color of Gluten-Free Muffin and All-Purpose Flour Muffin

The color of the gluten-free muffin and the all-purpose flour muffin were measured with a colorimeter (CR-400, Konica Minolta Sensing Inc., Osaka, Japan). Results were reported as color values in the CIE *L***a** *b** color space. The reported color of the muffin was the average color obtained from the measurement of the top and bottom surfaces of 3 dots on each side with 15 pieces per replication of each formulation.

#### 2.4.4. Firmness of the Gluten-Free Muffin and the All-Purpose Flour Muffin

The firmness of the gluten-free muffin and the all-purpose flour muffin were measured with Acquisition Rate (PRS) PRS probe surfaces (NEWTRY GY-4, Shanghai, China). The probe was pressed onto the sample at a velocity of 1 mm/s for a distance of 50% of the sample height. Then the probe was moved back up at a speed of 5 mm/s and the droplet traveled for 30 s. After that, the probe was pressed on the sample again at the same speed.

#### 2.4.5. Sensory Evaluation of the Gluten-Free Muffin and the All-Purpose Flour Muffin

Thirty panelists determined the sensory characteristics of the gluten-free muffin and the all-purpose flour muffin. Sensory attributes including color, taste, flavor, texture, and overall acceptance were evaluated using a 9-point hedonic scale, ranging from 1 (dislike extremely) to 9 (like extremely) [19,20].

### 2.5. Statistical Analysis

The experiments of proximate analysis, total cyanide content, acrylamide, amylose, gluten content, viscosity, bake loss, color of muffin, firmness of muffin, and sensory test were conducted in triplicate. Data reported are averages of three determinations, by dissolved one-way analysis of variance (ANOVA) using SPSS version 14. The New Duncan’s Multiple Range Test (DMRT) differentiated treatments at *p* ≤ 0.05.

## 3. Results

### 3.1. Proximate Analysis

The proximate composition of the cassava flour is shown in Table 1. The moisture content of three cultivars of cassava flour ranged between 10.56 ± 0.01 and 10.85 ± 0.45%, which is within the range acceptable for the effective storage of cassava flour. The total protein content of the flour shows that flour from three cultivars of cassava had a protein content of 1.97 ± 0.00%, 2.15 ± 0.01%, and 2.18 ± 0.01%, respectively. It was considered that the flour from three cultivars of cassava was the normal value of protein in cassava flour. There were no differences in the ash, total carbohydrate, and fat of the three cultivars of flour: in the ash 1.53 ± 0.07–1.91 ± 0.00%, carbohydrate 84.27 ± 0.04–84.91 ± 0.46%, and fat 0.79 ± 0.02–0.95 ± 0.01%.

### 3.2. Total Cyanide Content, Acrylamide, Amylose, Gluten Content, and Viscosity

The results of amylose content, acrylamide, cyanide content, gluten content, and viscosity of cassava flour revealed that amylose content at 19.93 ± 0.470%, and acrylamide and gluten content were not found in flour. The cassava flour’s cyanide content at 2.93 ± 0.16 mg/kg dry basis did not exceed the Codex Alimentarius limit (10 mg/kg of dry weight) considered harmless to consumers. The viscosity of the flour was 6286.00 ± 1.52 mPa.s (Table 2).

### 3.3. Color of Cassava Flour

Three varieties of cassava flour samples were evaluated in a colorimeter (CR-400, Konika Minolta), it was found that the color values (*L**, *a** and *b**) were non significantly different between samples. The characteristics of lightness *L** value were between 97.65 ± 0.65 and 99.46 ± 0.40 which represented the whiteness of the flour. The average *a** value (red–green degree) of flour samples was between 1.01 ± 0.34 and 1.75 ± 0.12. The flour derived from cassava varieties Pirun 4 had the highest redness. However, the three cassava varieties showed low redness in the flour. The *b** averaged (yellow–blue degree) between 14.73 ± 0.04 and 15.76 ± 1.20, indicating a slight yellowness of the flour. The overall color characteristics of each cassava flour are shown in Table 3.

### 3.4. FTIR Analysis

The comparison of the differences between the flour of three varieties of cassava by the FTIR technique shows the spectral characteristics of the flour of peaks in the range of 4000–800 cm^−1^ contained functional groups 3000–2800 cm^−1^, 1834–1583 cm^−1^, and 1200–800 cm^−1^, as shown in Figure 1A.

In addition, the principal components analysis (PCA) and loading plot analyzed the differences between the three varieties of cassava flour found that the three varieties were clearly different, as shown in Figure 1B. Loading was distinguished by the difference in the peak of cassava flour variety Pirun 6 (P6) in 2928, 1638, 1149, 1076, and 997. While Pirun 2 (P2) and Pirun 4 (P4) showed a dominant range of 1200–800, the spectrum in the Figure 1A shows the difference in the peak in the ranges 3000–2800 cm^−1^ and 1834–1583 cm^−1^ of flour from cassava, Pirun 6. Moreover, the cluster results showed that flour from P6 was the most different from the other two varieties, while P2 and P4 were closely related, giving results consistent with the PCA. The data analysis represents that the PC2 and PC3 regularly demonstrated the most different clustering of the three groups. The PC2 and PC3 loading of each cassava flour sample showed that the separation between PC2 and PC3 corresponded to a total variance of 29% from PC2 and 13% from PC3 (Figure 1B,C).

It was found that the cassava flour varieties Pirun 2 and Pirun 4 were similar in the cluster analysis for FTIR but were a different group compared with Pirun 6 (Figure 2). The results show the biochemicals accorded with the proximate composition of each cassava flour sample.

### 3.5. Microstructure of Cassava Flour

Microstructural analysis of flour samples using a scanning electron microscope (SEM) reveals that the morphology of different varieties of flour distribution of granules were found evenly. They were spherical and dome-shaped, ranging from 9 to 20 µm in size (Figure 3). There were no apparent differences between species with respect to granule morphology.

From the physico-chemical properties of three varieties of cassava flour, it was found that all of the flours were not different in terms of physico-chemical properties. Therefore, flour from cassava variety Pirun 6 was selected for use in the next experiments, because there was low cyanide and fat content, and high ash, protein, and viscosity values.

### 3.6. Effects on the Quality and Sensory Characteristics of the Gluten-Free Muffin

#### 3.6.1. The Physical Characteristics of the Gluten-Free Muffin

A basic muffin formula as compared with gluten-free muffins containing cassava flour was tested for baking loss (%), firmness (N), and color. The results found that gluten-free muffins had a higher bake weight loss than the basic muffin at 13.13 ± 0.56%, while the firmness values were not significantly different between the two muffin formulations (Figure 4). The color values found that gluten-free muffins had higher red (*a**) values than the basic muffin formula. However, the gluten-free muffins containing cassava flour were a yellow color (*b**) with no significant differences when compared with the basic muffins (Table 4).

#### 3.6.2. Sensory Evaluation of the Gluten-Free Muffin

The results of the sensory characteristics of the basic muffin formula and gluten-free muffins containing cassava flour showed that the texture appearance, color, taste, and crunchiness were not different. However, tasters rated the smell, wallowing sensation, and overall liking of gluten-free muffins containing cassava flour over basic muffins (Figure 5), indicating a sensory quality of gluten-free muffins comparable to that of the basic muffins.

## 4. Discussion

In Thailand, the local varieties of edible cassava used in the food industry have many variations in the nutrient quality of the cassava root depending on the cassava variety used. A study of the physico-chemical properties of flour from three varieties showed that the moisture content of three cultivars of cassava flour was between 10.56 ± 0.01 and 10.85 ± 0.45%. Hasmadi et al. [21] reported that high-quality cassava flour usually contains moisture content ranging from 6.34 to 14.58%. Moisture is an important parameter in the storage of cassava flour. A total protein content of 1.97 ± 0.00%, 2.15 ± 0.01%, and 2.18 ± 0.01%, respectively, was also found. Hasmadi et al. [21] and Peprah et al. [22] reported protein content of cassava flour in the range of 1–3% on a dry basis. Ash, total carbohydrates, and fat of the three cultivars of flour were different. Amylose content was 19.93 ± 0.470% and the viscosity was 6286.00 ± 1.52 times. According to the authors [23,24], the proximate analysis ranged from 8.79 to 9.35%, 0.55 to 26.23%, 0.34 to 2.01%, 0.32 to 8.24%, and 0.10 to 17.86% for moisture, protein, fat, and ash, respectively, while carbohydrate ranged from 36.31 to 89.62% and amylose contents were from 18.47 to 25.35%. Results of the color analyses of cassava flour samples found that the color values (*L**, *a** and *b**) were non-significantly different between samples. The characteristics of lightness *L** value were between 97.65 ± 0.65 and 99.46 ± 0.40 which represented the whiteness of the flour. The average *a** value (red-green degree) of flour samples was between 1.01 ± 0.34–1.75 ± 0.12. The *b** averaged (yellow–blue degree) between 14.73 ± 0.04 and 15.76 ± 1.20 indicated a slight yellowness of the flour. Chisenga et al. [8] reported that the difference in color in flour was due to the composition of the cassava such as ash content, protein, pigment, and starch. The color of the flour comes from the root flesh and the reaction of the complex with mucilage and latex as well as starch–lipid, fiber–lipid, and protein–lipid interactions. As a result, the cassava flour has an intense color [5,7]. The peak viscosity of flour from cassava Pirun 6 was 6286 ± 1.52 mPa.s, which was significantly higher than Pirun 4 (4640 ± 0.58 mPa.s) and Pirun 2 (3390 ± 0.57 mPa.s). The higher peak viscosity was related to a higher degree of swelling of the starch granules and also starch content [25]. The cluster analysis for the FTIR spectra of all samples found that the three varieties of flour were clearly different. Cassava varieties Pirun 6 had higher regions than flour from cassava varieties Pirun 2 and Pirun 4 at wave number 2928 cm^−1^ representing –C-H (CH_2_) stretching. The peak of the protein of flour from cassava varieties Pirun 6 was higher than Pirun 2 and Pirun 4 at wave number 1638 cm^−1^ representing amide I (-C=O) stretching, respectively. FTIR results showed proximate compositions of protein and fat content in flour from cassava varieties Pirun 6. The flour from cassava varieties Pirun 2, Pirun 4, and Pirun 6 quality traits varied among the cassava varieties, and even within the same shape, these physico-chemical components were still not consistent, and the source of variation was due to differences in flour fiber, ash, fat, protein, total cyanide, and peak viscosity. Therefore, it could be said that the differences in chemical components would be a characteristic of each variety [8,26]. Microstructural analysis of flour samples using a scanning electron microscope (SEM) revealed that the morphology of different varieties of flour distribution of granules was found evenly. They were spherical, dome-shaped, polyhedral, and truncated in shape from flour granules squeezed together ranging from 9 to 20 µm in size [8,27]. This is considered a typical standard size of cassava flour. The shape of the flour granules has not identified the difference between species with respect to granule morphology [28]. The test of physical properties and sensory attributes of gluten-free muffins compared with wheat flour found that gluten-free muffins had a higher bake weight loss than the basic muffin. That could be due to the limited moisture evaporation during baking by the enhanced water binding capacity [29]. The color values found that gluten-free muffins had higher red (*a**) values than the basic muffin formula. This is because cassava flour affects the formation of color in gluten-free muffins such as the Maillard reaction (reaction between carbohydrates with protein in the product) and caramelization (the sugar contained in the product is exposed to heat) at the time of baking [30]. From this experiment, it can be concluded that the gluten-free muffins had overall preferences similar to basic muffins.

## 5. Conclusions

Edible cassava of the three varieties—Pirun 2, Pirun 4, and Pirun 6—eventually have the potential to be applied in the food industry. Several analyses of cassava flour’s physico-chemical suitability and quality of flour for gluten-free muffins (functional properties) showed positive results. The results found that different varieties influenced the proximate compositions of cassava flour, and significant differences were observed in ash, fat, protein, total cyanide, and peak viscosity. Significant differences were also observed for the spectral characteristics of the flour from cassava variety Pirun 6 showed chemical components (functional groups) such as lipids, proteins, etc., by the FTIR technique. The physical characteristics of gluten-free muffins showed that the characteristics and color values of muffins using cassava flour had no significant difference when compared with muffins using wheat flour. The sensory evaluation of gluten-free muffins using cassava flour compared with wheat flour was acceptable by the sensory panelists. These studies may contribute to the development of gluten-free products based on cassava flour, aiming to substitute for wheat flour.

## Figures and Tables

**Figure 1 foods-11-04053-f001:**
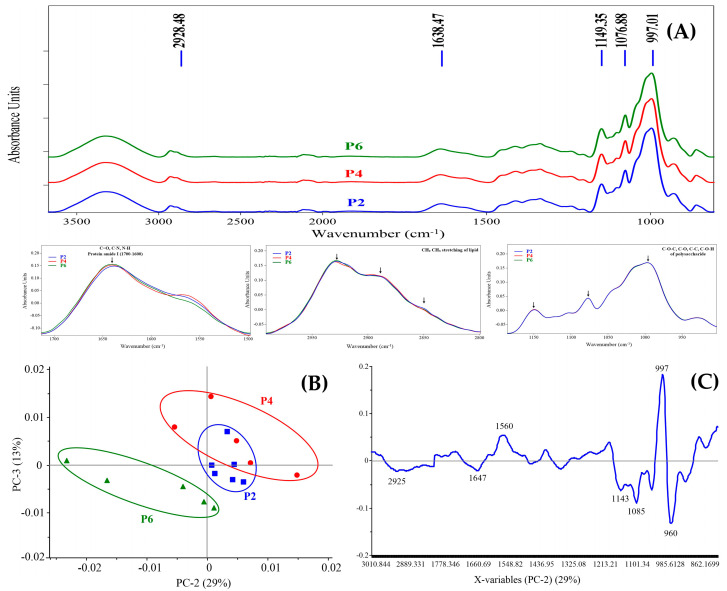
Comparison of three varieties of cassava flour by FTIR technique (**A**) average absorbance spectra (**B**) PCA analysis (**C**) loading plot.

**Figure 2 foods-11-04053-f002:**
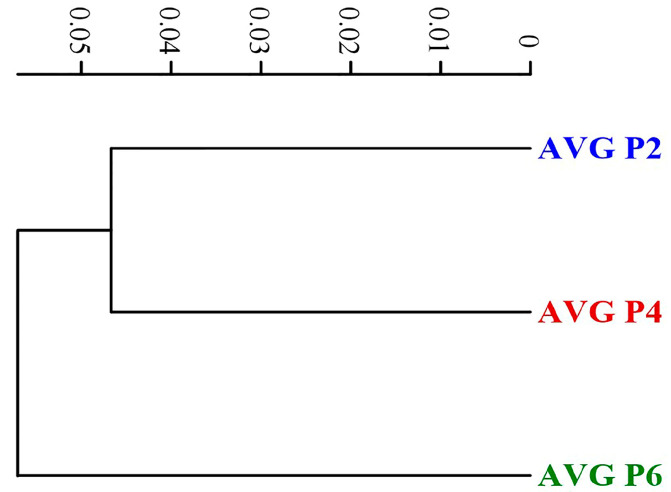
Cluster analysis for FTIR spectra compares three varieties of cassava flour (P2 = Pirun 2, P4 = Pirun 4, P6 = Pirun 6).

**Figure 3 foods-11-04053-f003:**
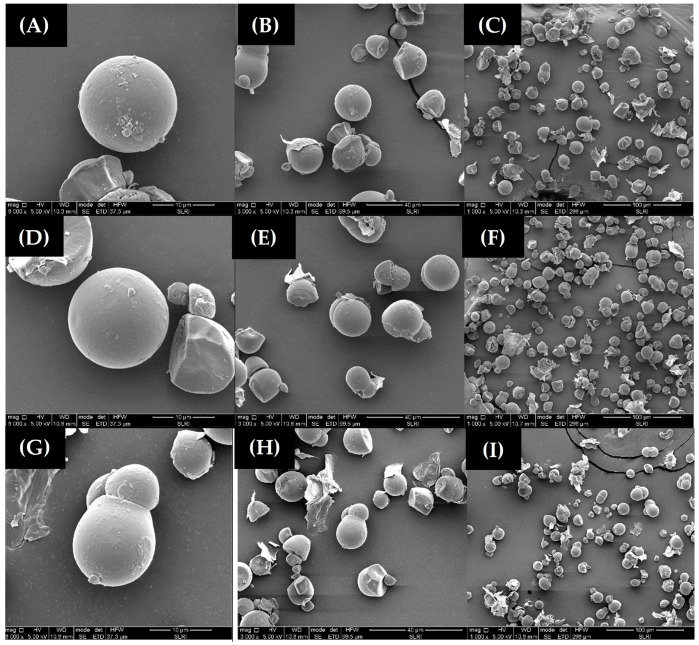
Characteristics of cassava flour granules, Pirun 6 (**A**–**C**), Pirun 4 (**D**–**F**), Pirun 2 (**G**–**I**) were photographed under a scanning electron microscope (SEM) at 1000× (**C**,**F**,**I**) 3000× (**B**,**E**,**H**) and 8000× (**A**,**D**,**G**).

**Figure 4 foods-11-04053-f004:**
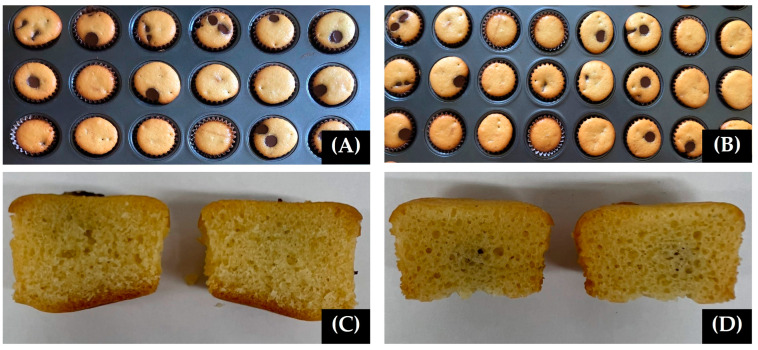
Characteristics of muffins. Muffins using wheat flour (**A**,**C**) and muffins using cassava flour (**B**,**D**).

**Figure 5 foods-11-04053-f005:**
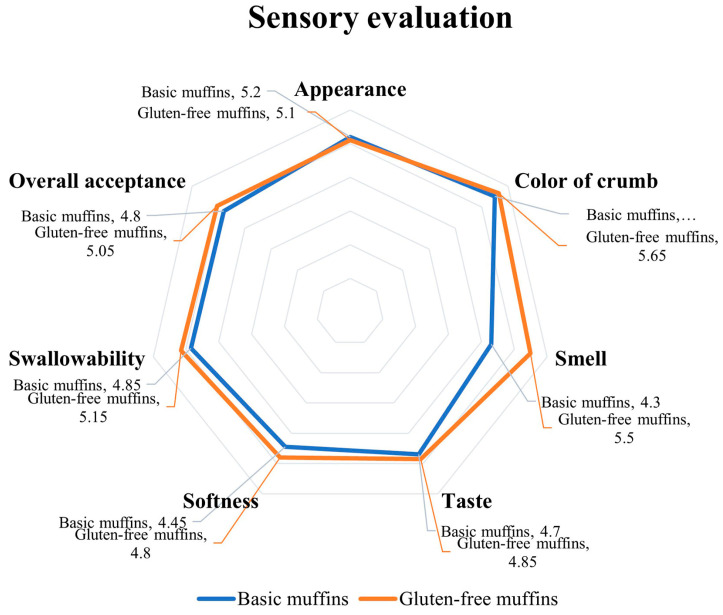
Sensory evaluation values of gluten-free muffins from cassava flour.

**Table 1 foods-11-04053-t001:** Proximate analysis of flour from difference varieties of cassava flour.

Cassava Flour Varieties	Proximate ^1/, 2/^				
	Ash (%)	Carbohydrate (%)	Fat (%)	Moisture (%)	Protein (%)
Flour from Pirun 2	1.53 ± 0.07 ^c^	84.32 ± 0.11	0.95 ± 0.01 ^a^	10.65 ± 0.01	2.15 ± 0.01 ^a^
Flour from Pirun 4	1.69 ± 0.02 ^b^	84.91 ± 0.46	0.87 ± 0.00 ^b^	10.56 ± 0.01	1.97 ± 0.00 ^b^
Flour from Pirun 6	1.91 ± 0.00 ^a^	84.27 ± 0.04	0.79 ± 0.02 ^c^	10.85 ± 0.45	2.18 ± 0.01 ^a^
F-Test	**	ns	**	ns	**

^1/^ Mean of percent dry substance (%) ± Standard deviation. ^2/^ Different letters in the same column represent significant differences, ns = not statistically significantly different. *, ** = statistically significant difference *p* ≤ 0.05 and highly significant *p* ≤ 0.01, respectively.

**Table 2 foods-11-04053-t002:** Amylose content, acrylamide, cyanide content, and viscosity from difference varieties of cassava flour.

Cassava Flour Varieties	Amylose Content ^1/^ (%)	Acrylamide ^2/^ (µg/kg)	Total Cyanide ^1/, 3/^ (mg/kg Dry Basis)	Gluten Content ^2/^ (mg/kg Dry Basis)	Peak Viscosity ^1/, 3/^ (mPa.s)
Flour from Pirun 2	19.10 ± 0.63	N	3.17 ± 0.44 ^b^	N	3390.00 ± 0.57 ^c^
Flour from Pirun 4	20.30 ± 0.74	N	5.68 ± 0.51 ^a^	N	4640.00 ± 0.58 ^b^
Flour from Pirun 6	19.93 ± 0.47	N	2.93 ± 0.16 ^b^	N	6286.00 ± 1.52 ^a^
F-Test ^3/^	ns	-	**	-	**

^1/^ Mean ± Standard deviation; ^2/^ N = Not Detected. ^3/^ Different letters in the same column represent significant differences, ns = not statistically significantly different. *, ** = statistically significant difference *p* ≤ 0.05 and highly significant *p* ≤ 0.01, respectively.

**Table 3 foods-11-04053-t003:** Color parameters of flour from three varieties of cassava.

Cassava Flour Varieties	Color Parameter^1/^		
	*L**	*a**	*b**
Flour from Pirun 2	98.30 ± 2.77	1.15 ± 0.37	15.76 ± 1.20
Flour from Pirun 4	97.65 ± 0.65	1.75 ± 0.12	15.51 ± 0.31
Flour from Pirun 6	99.46 ± 0.40	1.01 ± 0.34	14.73 ± 0.04
F-Test ^2/^	ns	ns	ns

^1/^ Mean ± Standard deviation; * The CIE = *L**, lightness; *a**, redness; *b**, yellowness. ^2/^ ns = not statistically significantly different. *, ** = statistically significant difference *p* ≤ 0.05 and highly significant *p* ≤ 0.01, respectively.

**Table 4 foods-11-04053-t004:** Color values and physical characteristics of gluten-free muffins from cassava flour.

Treatments ^1/^	Bake loss (%)	Firmness (N)	Color ^2/^		
			*L**	*a**	*b**
Basic muffins	10.27 ± 1.29	07.72 ± 0.53	67.46 ± 2.06	11.18 ± 1.33 ^b^	42.17 ± 0.43
Gluten-free muffins	13.13 ± 0.56	07.22 ± 0.65	57.99 ± 2.88	20.79 ± 2.39 ^a^	42.99 ± 1.56
F-Test	ns	ns	ns	*	ns

^1/^ Mean ± Standard deviation; * The CIE = *L**, lightness; *a**, redness; *b**, yellowness. ^2/^ Different letters in the same column represent significant differences, ns = not statistically significantly different. *, ** = statistically significant difference *p* ≤ 0.05 and highly significant *p* ≤ 0.01, respectively.

## Data Availability

The data used to support the findings of this study can be made available by the corresponding author upon request.

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
