# Peer review of "Flour on Gluten-Free Muffins from Different Edible Cassava Varieties in Thailand"

_foods, 2022, doi:10.3390/foods11244053_

Round 1
Reviewer 1 Report
In this paper, suitability and quality of flour produced from different edible cassava varieties assessed. However, some modifications should be revised.
1. Lines:19 and 166-167. The author explained in the abstract that there was no significant difference in the protein content of the three cassava flours, but pointed out that there was a significant difference in the protein content between the three cassava flours when the infrared results were analyzed. Is the article logic reasonable? Please check.
2. Table 2. Please do a significant difference analysis of the data in the table.
3. Lines:182-183. The paper pointed out that the cluster analysis of FTIR spectra of all flour samples showed that the differences among all three flour varieties were clearly distinguished and consistent with PCA. Lines:184-185. The results show biochemical accorded with the proximate composition of each cassava flour sample.
Lines:279-280. This study found all the varieties of cassava were similar physico-chemical properties. There is a logical conflict between these three parts, please explain the specific differences between the three cassava flour.
4. Lines: 259-260. It was pointed out that there were differences in flour quality among cassava varieties, which were resulted from the differences in flour particle size, fiber and ash content. However, from the results of ash content, infrared and scanning electron microscopy, there was no significant difference in quality shape between the three cassava flours. In addition to viscosity, please provide additional data indicators to highlight the essential difference between the cassava variety Pirun 6 and the other two cassava flours.
Reviewer 2 Report
Dear Editors,
In my opinion, the reviewed manuscript “Evaluation of Suitability of Different Edible Cassava Varieties for Flour Production in Thailand” needs a major rewrite.
Suggested changes are included in the review addressed to the authors.
Kind regards
Dear Authors,
In my opinion, the manuscript "Evaluation of Suitability of Different Edible Cassava Varieties for Flour Production in Thailand" submitted for review needs a thorough rewrite.
1. The title and purpose of the work are not fully consistent with the scope of the research. I think that muffins should be included in the title. The purpose of the work also requires redrafting.
2. In point 2.1. please provide information on whether the cassava came from a controlled field trial or industrial cultivation. What agricultural techniques were used during cultivation? What were the weather conditions in the 2021 growing season?
3. In point 2.2. please specify in which dryer the cassava drying process was carried out and on what device the grinding process was carried out.
4. In point 2.3. please specify the methodology used to assess the chemical composition of cassava flour. Please specify what methods and devices were used to perform the individual determinations, in a way that allows the experiment to be reproduced. References should be supplemented with AOAC items …., and in the text of the manuscript (lines 77, 82) give a reference to the number of these references in References.
5. In point 2.4. please provide information on the equipment used to prepare the dough and bake muffins. Information on what constituted the control sample should also be provided. Titles 2.4.2, 2.4.3, and 2.4.4 require redrafting, the current wording suggests that only gluten-free muffins were baked, while the data in Table 3 shows that the baking loss, hardness, and color of wheat flour muffins were baked and evaluated.
6. In point 2.5. not all methods used for the statistical evaluation of results are given, please complete them.
7. Chapter 3. It is not entirely clear whether the content of chemical components in cassava flour (Table 1) is given in % or terms of dry substance. I have a question, has an assessment of the chemical composition of wheat flour used as a raw material for baking muffins as a control sample been carried out? I believe these results should be included in the manuscript.
8. In point 3.3. no table with the results of the assessment of flour color parameters described in the text of the manuscript. Table 3 with the results of the evaluation of the physical characteristics and color of the muffins should be transferred to point 3.6.1.
9. Chapter 4 needs a major rewrite. In its current form, it mostly duplicates the information provided in Chapter 3. It is not fully understood why the gluten-free muffins had a higher proportion of red color than the control muffins made of wheat flour. Wheat flour also contains carbohydrates and several times more protein than cassava flour, and Maillard reactions also occur during baking.
Please remove the year of publication (lines 233, 248) instead of the year and insert reference numbers from References (line 236).
10. The summary needs a major rewrite
11. The bibliography contains only 22 items, it needs to be supplemented, and it is necessary to extend the discussion of the results.
Best regards
Reviewer 3 Report
revising it carfully and fixing the following points:
- The point is not novel enough, and the author should critically explain what the novel point they are going to add for the scientific community, I insist for this point, otherwise, i suggest to reject it.
- Explain the difference between your study and the other studies like: https://www.academia.edu/59196967/Assessment_of_the_Suitability_of_Different_Cassava_Varieties_for_Gari_and_Fufu_Flour_Production_in_Liberia
- The figures quality needs to be improved, especially the visual observation of the muffin they produced.
- There is no stability study of the muffin produced, and this is very important to study the storage stability of the end product.
- FTIR results explination are not good enough and more disuccion and related examples are still needed.
- Figure 1 no need to include, as it is not representive for anything.
- The language is really not clear, and more efforts are needed.
Round 2
Reviewer 1 Report
The authors have revised their manuscript carefully
Reviewer 3 Report
It could be accepted in its current form.